# The Application of Hyperspectral Remote Sensing Imagery (HRSI) for Weed Detection Analysis in Rice Fields: A Review

Nursyazyla Sulaiman [1], Nik Norasma Che'Ya [1,*], Muhammad Huzaifah Mohd Roslim [2], Abdul Shukor Juraimi [3], Nisfariza Mohd Noor [4] and Wan Fazilah Fazlil Ilahi [1]

[1] Department of Agriculture Technology, Faculty of Agriculture, Universiti Putra Malaysia (UPM), Serdang 43400, Malaysia; sya_zyla@yahoo.com (N.S.); wanfazilah@upm.edu.my (W.F.F.I.)

[2] Department of Crop Science, Faculty of Agricultural Science and Forestry, Universiti Putra Malaysia Bintulu Campus, Bintulu 97000, Malaysia; muhammadhuzaifah@upm.edu.my

[3] Department of Crop Science, Faculty of Agriculture, Universiti Putra Malaysia (UPM), Serdang 43400, Malaysia; ashukur@upm.edu.my

[4] Department of Geography, Faculty of Arts and Social Sciences, University of Malaya, Kuala Lumpur 50603, Malaysia; nish@um.edu.my

* Correspondence: niknorasma@upm.edu.my; Tel.: +603-9769-4892

**Abstract:** Weeds are found on every cropland across the world. Weeds compete for light, water, and nutrients with attractive plants, introduce illnesses or viruses, and attract harmful insects and pests, resulting in yield loss. New weed detection technologies have been developed in recent years to increase weed detection speed and accuracy, resolving the contradiction between the goals of enhancing soil health and achieving sufficient weed control for profitable farming. In recent years, a variety of platforms, such as satellites, airplanes, unmanned aerial vehicles (UAVs), and close-range platforms, have become more commonly available for gathering hyperspectral images with varying spatial, temporal, and spectral resolutions. Plants must be divided into crops and weeds based on their species for successful weed detection. Therefore, hyperspectral image categorization also has become popular since the development of hyperspectral image technology. Unmanned aerial vehicle (UAV) hyperspectral imaging techniques have recently emerged as a valuable tool in agricultural remote sensing, with tremendous promise for weed detection and species separation. Hence, this paper will review the weeds problem in rice fields in Malaysia and focus on the application of hyperspectral remote sensing imagery (HRSI) for weed detection with algorithms and modelling employed for weeds discrimination analysis.

**Keywords:** rice plant; weed; hyperspectral imagery; remote sensing

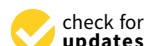

## 1. Introduction

The agricultural sector provides significant economic growth by endowing food sources, producing industrial raw materials as well as providing job opportunities for a substantial number of individuals [1,2]. In Malaysia, the agricultural industry has endured as one of the predominant sectors for socio-economic activity, contributing about 8.7% of the annual gross domestic product (GDP) and 11.4% of the total employment [3]. The major agricultural activities in Malaysia are dominated by rubber (*Hevea brasiliensis* (Willd. Ex A. Juss) *Mull. ARg*), oil palm (*Elaeis guineensis*) and rice plant (*Oryza sativa* L.) [4]. The agricultural sector focuses on sustainable food production and proffering consistent, high-quality and safe food products. In line with an increasing population, global food production will need to significantly multiply in the next few years along with limited area expansion [5]. However, there are several issues that have arisen regarding low crop yield production such as uncertain weather conditions, insufficient labour power, unmaintained agricultural instruments, a reduction in soil and seed quality, constraints on the use of new technologies, etc. [4,6].

In the agricultural ecosystem, weeds serve as a competitor with the actual crop for obtaining light source, nutrients, moisture intensity and gaseous exchange which result in a reduction in crop yield and product quality [3,7]. For crop production, the potential of weed-induced deprivation refers to the type of weed, density, emergence time, and duration intrusion including the simultaneous emergence of weeds along with crop-augmented competition towards restricted growth resources that can trigger the risk of critical yield loss [2,8]. This paper will review the weeds problem in rice fields in Malaysia and focus on the application of hyperspectral remote sensing imagery (HRSI) for weed detection analysis. As a result, researchers, particularly in developing nations, can apply their understanding of decreasing weed presence and enhancing yield output. The focus of this work is on weed detection in rice fields utilising a hyperspectral remote sensing platform. However, hyperspectral remote sensing weed detection in other crops is also included in this review.

## 2. Methodology

This paper is a conventional review paper. Sources of articles and related research papers were browsed and identified from several databases such as Google Scholar, Google Book, Semantic Scholar, UPM EZAccess, MDPI, and ResearchGate. The primary keyword 'hyperspectral remote sensing' and its synonym paired with the secondary keyword 'weed' and the third keyword 'rice plant' was used as the source of content exploration. Each database search made use of these keyword sets. A hand search was also performed to ensure that no related articles were overlooked. The search was carried out in the fourth quarter of 2021.

All search results were filtered using the following criteria: (1) the study must use hyperspectral remote sensing imagery and platform as the primary data input, (2) the study must dispute the application of hyperspectral remote sensing techniques in weed detection analysis, (3) the document must have reported on the research undertaken, (4) the included papers must have been published in the first quarter of 2021, and (5) the articles must be written in English.

The articles were then reviewed by title and abstract to exclude those that did not satisfy the requirements. Finally, the complete text of the remaining articles was scrutinised to determine whether or not they fit the requirements. Finally, data from a number of articles were taken and transferred into a spreadsheet. Citation information, study objectives, hyperspectral remote sensing sensor, crop and weed types, methodologies and techniques employed, accuracy evaluation, study implications, year of publication, and reference data were all included in the details.

This paper is organized into seven sections. The first section describes the weeds problem in agricultural crops. The approach for searching the scientific database for relevant publications is explained in Section 2. Section 3 highlights the significance of weeds presence in Malaysia's rice fields. Section 4 elaborates on the hyperspectral remote sensing system while the literature on several methodologies for handling remote sensing datasets is presented in Section 5. Section 6 explains the weed detection analysis by using hyperspectral remote sensing with the classification by using spectral reflectance and utilization of the modelling and algorithm. Future directions of hyperspectral remote sensing approaches and conclusions are presented in Section 7.

## 3. Weeds in Malaysia's Rice Field

According to El Pebrian and Ismail [9], rice is one of Malaysia's most widely grown agricultural crops. According to the Department of Agriculture (DoA) Peninsular Malaysia, this crop was grown on 679,239 acres in 2014, making it the third-largest crop in the country after oil palm and rubber. Malaysia produced 2,848,559 metric tonnes of paddy with such a large planted area. Yusof et al. [10] stated that Malaysian farmers produced 70% of the country's rice production while the rice industry's role is not only to contribute to Malaysia's economy but also to ensure the country's food security. The plantation of rice occurs twice a year, for example: (i) Main season (October–March) and off season

(April–September) with two types of methods implemented in rice cultivation known as direct seeding and transplanting. Dilipkumar et al. [4] stated that 90% of the total rice crop in Malaysia was planted using the direct-seeded method and 10% was transplanted.

Most farmers chose the direct seeding method due to labour insufficiency and expensive cost in rice transplanting [11]. Direct-seeded rice systems implement three different principal methods, for example, water seeding, wet seeding and dry seeding [4,12]. However, the substitution of this planting technique presents crucial weeds expansion in rice crops. According to Hossain et al. [13], many studies have reported that the dominance of particular weed species in rice cropping systems is significantly influenced by the crop establishment method. According to Nagarde et al. [14], weeds are a severe danger to rice, with yearly weed yield losses ranging from 15% to 21% worldwide. Due to massive weed infestation, direct-seeded rice yields are predicted to be reduced by 60% and above. Yield reductions of up to 48% in transplanted crops, 53% in direct-seeded crops (flooded conditions) and 74% in direct-seeded crops (dry soils) have been documented. Different types of grasses, broadleaf weed and sedges make up the weed flora in direct-seeded rice (Table 1).

**Table 1.** Weeds species in Asia's rice field.

| Grassy Weeds | Sedges | Broadleaf Weeds |
|---|---|---|
| *Digitaria setigera* | *Cyperus iria* | *Commelina benghalensis* |
| *Digitaria sanguinalis* | *Cyperus difformis* | *Caesulia axillaris* |
| *Digitaria ciliaris* | *Cyperus rotundus* | *Eclipta prostrata* |
| *Echinochloa colonum* | *Fimbristylis miliacea* | *Ipomoea aquatica* |
| *Echinoclhoa crus-galli* | | *Ludwigia octovalvis* |
| *Eleusine indica* | | *Ludwigia adscendens* |
| *Ischaemum rugosum* | | *Monochoria vaginalis* |
| *Leptochloa chinensis* | | *Sphenoclea zeylanica* |
| *Oryza sativa* | | |
| *Paspalum* | | |

Source: Nagarde et al. [14]

Direct-seeded rice systems bestow an aerobic environment for weed growth since they are not flooded during the beginning growth stage of the rice plants and it is convenient towards weed expansion [11,15]. The aerobic soil condition in the direct-seeded rice system conserves water, while the weed problem in direct-seeded rice is exacerbated by the lack of stagnant water and the lack of a 'head start' in rice seedlings over sprouting weed seedlings [16]. Toriyama [17] explained that the extensive employment of the direct seeding method with the frequent use of herbicide and a shortage of irrigation supplies accountable on the transference of weed species populations in the rice field ecosystem, for example, the grasses species: *Echinochloa crus-galli, Echinochloa* spp. *(E. oryzicola, E. colona, E. staginina, and E. picta), Leptochloa Chinensis,* and *Ischaemum rugosum*, which were not dominant in Malaysian rice fields, has previously become widespread afterwards (see Table 2). Furthermore, Chauhan et al. [12] found that the density of grassy weeds in zero-tilled direct-seeded rice was higher than in puddled transplanted rice. Sedges and broadleaves, on the other hand, were less abundant. Broadleaves such as *Sagittaria guayanensis* Kunth, *Monochoria vaginalis* (Burm. f.) C. Presl ex Kunth, *Limnocharis flava* (L.) *Buchenau, Ludwigia octovalvis* (Jacq.) P.H. Raven, and *Alternanthera sessilis* (L.) R. Br. Ex DC. and *Ammannia baccifera* L. also had expanded abundance in the puddled transplanted field.

**Table 2.** Weed shift from transplanting to the direct-seeding method.

| Irrigated Transplanting | Extensive Direct Seeding | Intensive Direct Seeding |
|:---:|:---:|:---:|
| **Grasses** | | |
| | *Echinochloa crus-galli complex* | *E. crus-galli* |
| *Isachne globose* | *Lepthochola chinensis* | *L. chinensis* |
| *Leersia hexandra* | *Ischaemum rugosum* | *I. Rugosum* |
| | *Oryza sativa* (weedy rice) | *O. sativa* (weedy rice) |
| **Broadleaf weeds** | | |
| | | *L. flava* [a] |
| *Limnocharis flava* | *L. flava* | *S. guyanensis* [b] |
| *Monochoria vaginalis* | *M. vaginalis* | *Sphenoclea zeylanica* [b] |
| *M. minuta* | *L. hyssopifolia* | *M. crenata* [b] |
| *Sphenoclea zeylanica* | *M. minuta* | *Limnophila erecta* [a] |
| **Sedges** | | |
| | | *C. iria* |
| *Scirpus grossus* | *Cyperus iria* | *F. milicea* [b] |
| | *Fimbristylis miliacea* | *C. difformis* |

[a] Biotypes with herbicide resistance against 2,4-D and ALS-inhibitor herbicides. [b] Species/biotypes with herbicide resistance against 2,4-D. Source: [17].

Weeds are one of the most significant causes of reducing rice productivity, resulting in not only large financial expenditures but also crop quality difficulties. Crops can also be affected by weeds present at any growth stage [18]. In Malaysia, weed-related production losses range from 5% to 85%, depending on the planting method, season, region, major weed flora, weed density, management practices and infestation length [4]. Issues regarding weeds in crops are complex; meanwhile, to reduce their expansion and impacts on the crop, the management strategy chosen must be synchronized in all aspects to make sure that systematic guidance will be assembled to manage the existing weeds as well as to prevent the spreading of new weeds [19]. A particular weed management proposition, for example, mechanical, chemical, manual and biological control strategies were initiated for weeds control in a crop field since these strategies came with certain constraints such as proper climatic circumstances, location of farmers, labour availability and the capability to endure with management expenses [20,21]. Early weed treatment not only reduces the occurrence of pests and diseases but also reduces agricultural yield loss by up to 34%. Chemical and non-chemical weed management strategies have been widely used in rice fields in this scenario. Manual weeding is too time consuming, expensive, and inconvenient as a non-chemical technique. Mechanical weed management is a non-chemical approach [18].

Partel et al. [22] created and constructed a smart sprayer that could distinguish between weeds and non-weed objects using machine vision and artificial intelligence. This targeted approach was combined with a revolutionary precision spraying system that included a state-of-the-art weed detecting technology and a weed mapping system for precise spraying. When compared to traditional broadcast spraying techniques, which often cover the entire field, the results showed that using this system lowered the number of agrochemicals necessary. Huang et al. [23] and Yao and Huang [24] mentioned that agricultural remote sensing has been established and utilised for monitoring crop field conditions such as growth status, soil variability, crops stress from weeds, pests, water and nutrition insufficiency in providing data and information towards the efficient operation. Unmanned aerial vehicle (UAV) technology provides a desirable precision agriculture data gathering platform that is highly flexible and simple to use while collecting high spatial resolution data in a timely way. Due to their geographical and temporal resolution capabilities and cost-effectiveness, UAVs are a better platform for crop monitoring activity [25]. Currently, in conjunction with the evolving transducer technology and sensor, remote sensing approaches were upgraded for weed detection and control particularly with the emergence of hyperspectral sensing and imaging [23].

## 4. Hyperspectral Remote Sensing: A Brief Overview

According to Weiss et al. [26], agriculture monitoring from remote sensing is a vast subject that has been widely addressed from multiple perspectives, sometimes based on specific applications (e.g., precision farming, yield prediction, irrigation, weed detection), remote sensing platforms (e.g., satellites, unmanned aerial vehicles—UAVs, unmanned ground vehicles—UGVs), or sensors (e.g., active or passive sensing, wavelength domain) or specific locations and climatic contexts (e.g., country or continent, wetlands or drylands). Campbell and Wynne [27] defined remote sensing as the application of acquiring information regarding the Earth's land and water surface by utilising images obtained from an overhead perspective, implementing electromagnetic radiation in one or more regions of the electromagnetic spectrum, reflected or emitted from the Earth's surface. Hyperspectral remote sensing involves extracting information from the objects or scenes that lie on the Earth's surface due to radiance obtained by airborne or spaceborne sensors [28,29].

Generally, hyperspectral imaging is an incorporation of the modern imaging system and traditional spectroscopy technology [30,31]. According to Govender et al. [32], the evolution of airborne and satellite hyperspectral sensor technologies has overcome the restraint of multispectral sensors since hyperspectral sensors assemble several narrow spectral bands from the visible, near-infrared (NIR), mid-infrared, and short-wave infrared portions of the electromagnetic spectrum. The hyperspectral sensor collects about 200 or more spectral bands, each only 10 nm wide [27] which allows the construction of continuous spectral reflectance signatures while the narrow bandwidths element of hyperspectral data enable in-depth examination of Earth surface characteristics which would disappear within the relatively coarse bandwidths acquired with multispectral data. Hyperspectral data are usually assigned as hypercubes (see Figure 1) that contain two spatial dimensions and one spectral dimension, regarding the characteristics of each hyperspectral image, comprising many channels since there were bands—in contrast to grayscale or RGB images—that included only one or three channels, respectively [33].

The hyperspectral data cube in Figure 1 explained that Figure 1a A push-broom sensor on an airborne or spaceborne platform acquire spectral data for a one-dimensional row of cross-track pixels named as scanline; Figure 1b Sequential scan lines including spectra for each row of cross-track pixels are pilled to obtain a three-dimensional hyperspectral data cube which in this illustration the spatial details of a scene are constituted by the x and y dimensions of the cube, while the amplitude spectra of the pixels are projected to the z dimension; Figure 1c the three-dimensional hyperspectral data cube can be analysed as a stack of two-dimensional spatial images whereas each is equivalent to a particular narrow waveband. Usually, hyperspectral data cubes contain hundreds of stacked images; Figure 1d the spectral samples can be marked for each pixel and discrimination of the features in the spectra deliver the primary mechanism for detection and classification in a scene [34,35]. Qian [31] stated that there were about three different methods in obtaining the hyperspectral data regarding the type of imaging spectrometers such as dispersive elements-based approach, spectral filters-based approach and snapshot hyperspectral imaging. In order to collect the hyperspectral images with different spatial and temporal resolutions, the sensors used can, for example, be mounted on different platforms. Unmanned-aerial vehicles (UAVs), airplanes, and close-range platforms [36]. Table 3 shows the comparison of different types of hyperspectral imaging platforms. Kate et al. [37] mentioned that hyperspectral sensors were utilised for providing information such as airborne visible/infrared imaging spectrometer (AVIRIS), Hyperion, Hymap (from HyVista Castle Hill, Australia), and airborne imaging spectroradiometer for applications (AISA). Table 4 below shows different types of hyperspectral sensors used which are usually mounted on the aircraft and satellite [38].

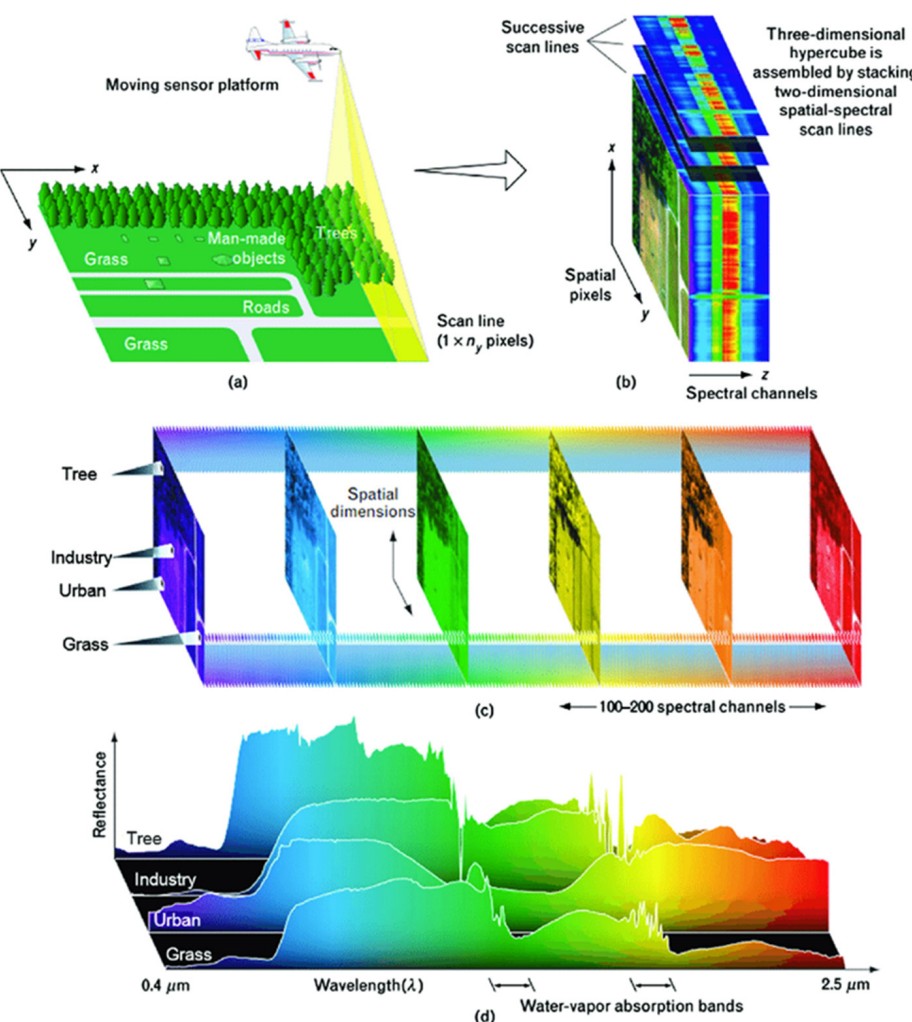

**Figure 1.** Hyperspectral data cube structure [34,35].

**Table 3.** Comparison of hyperspectral imaging platforms [36].

| Parameters | Satellites | Airplanes | Helicopters | Fixed-Wing UAVs | Multi-Rotor UAVs | Close-Range Platforms |
|---|---|---|---|---|---|---|
| Operational Altitudes | 400–700 km | 1–20 km | 100 m–2 km | <150 m | <150 m | <10 m |
| Spatial coverage | 42 km × 7.7 km | ~100 km$^2$ | ~10 km$^2$ | ~5 km$^2$ | ~0.5 km$^2$ | ~0.005 km$^2$ |
| Spatial resolution | 20–60 m | 1–20 m | 0.1–1 m | 0.01–0.5 m | 0.01–0.5 m | 0.0001–0.01 m |
| Temporal resolution | Days to weeks | Depends on flight operations (hours to days) | | | | |
| Flexibility | Low (fixed by repeating cycles) | Medium (depend on availability of aviation company) | | High | | |
| Operational complexity | Low (provide final data to users) | Medium (depend on users or vendors) | | High (operate by users with setting up the hardware and software) | | |
| Applicable scales | Regional–global | Landscape-regional | | Canopy–landscape | | Leaf–canopy |
| Major limiting factors | Weathers | Unfavourable flight height/speed, unstable illumination conditions | | Short battery endurance, flight regulations | | Platform design and operation |
| Image acquisition cost | Low to medium | High (typically need to hire an aviation company to fly) | | Large (due to area coverage) | | |

**Table 4.** Type of hyperspectral sensors on aircraft and satellites [38].

| Types of Sensors | Producer | Number of Bands | Spectral Image (μm) |
|---|---|---|---|
| **Satellite mounted hyperspectral sensors** | | | |
| FTHSI on MightySat II | Air Force Research (OH, USA) | 256 | 0.35–1.05 |
| Hyperion on EO- | NASA Guddard Space Flight Center (Greenbelt, MA, USA) | 242 | 0.40–250 |
| **Aircraft-mounted hyperspectral sensors** | | | |
| AVIRIS (airborne visible infrared imaging spectrometer) | NASA Jet Propulsion Lab. (Pasadena, CA, USA) | 224 | 0.40–2.50 |
| HYDICE (hyperspectral digital imagery collection experiment) | Naval Research Lab (Washington, DC, USA) | 210 | 0.40–2.50 |
| PROBE-1 | Earth Search Sciences Inc. (Kalispell, MT, USA) | 128 | 0.40–2.50 |
| CASI (compact airborne spectrographic imager) | ITRES Research Limited (Calgary, AB, Canada) | Over 22 | 0.40–1.00 |
| HyMap | Integrated Spectronics | 100 la 200 | Visible to thermal Infrared |
| EPS-H (environmental protection system) | GER Corporation | VIS/NIR (76), SWIR1 (32), SWIR2 (32), TIR (12) | VIS/NIR (0.43–1.05) SWIR1 (1.50–1.80) SWIR2 (2.00–2.50) TIR (8–12.50) |
| DAIS 7915 (digital airborne imaging spectrometer) | GER Corporation (geophysical and environmental research imaging spectrometer) | VIS/NIR (32), SWIR1 (8), SWIR2 (32), MIR (1), TIR (12) | VIS/NIR (0.43–1.05) SWIR1 (1.50–1.80) SWIR2 (2.00–2.50) MIR (3.00–5.00) TIR (8.70–12.30) |
| DAIS 21115 (digital airborne imaging spectrometer) | GER Corporation | VIS/NIR (76), SWIR1 (64), SWIR2 (64), MIR (1), TIR (6) | VIS/NIR (0.40–1.00) SWIR1 (1.00–1.80) SWIR2 (2.00–2.50) MIR (3.00–5.00) TIR (8.00–12.00) |
| AISA (airborne imaging spectrometer) | Spectral Imaging | Over 288 | 0.43–1.00 |

## 5. Hyperspectral Remote Sensing Imagery (HRSI) Data Processing and Analysing

### 5.1. Data Preprocessing

According to Weng and Xiaofei [39], due to the high-dimensional nature of hyperspectral data, as well as the resemblance between the spectra and mixed pixels, hyperspectral image technology still confronts a number of issues, the most pressing of which are the following: (1) Hyperspectral image data have high dimensionality. Because hyperspectral images are created by combining hundreds of bands of spectral reflectance data gathered by airborne or space-borne imaging spectrometers, the spectrum information dimension of hyperspectral images can also be hundreds of dimensions; (2) missing labelled samples. In practical applications, collecting hyperspectral image data is rather simple, but obtaining image-like label information is quite challenging. As a result, the categorization of hyperspectral pictures is sometimes hampered by a shortage of labelled samples; (3) variability

in spectral information across space. The spectral information of hyperspectral images changes in the spatial dimension as a result of factors such as atmospheric conditions, sensors, the composition and distribution of ground features, and the surrounding environment, resulting in the ground feature corresponding to each pixel not being single; and lastly (4) image quality which is the interference of noise and background elements during the acquisition of hyperspectral pictures which has a significant impact on the quality of the data collected. The categorization accuracy of hyperspectral images is directly influenced by the image quality.

Hyperspectral images obtained by various platforms and sensors are usually presented in raw format which requires them to be pre-processed (for example, atmospheric, radiometric, and spectral corrections) to rectify detailed information [36]. Assembling hyperspectral data is more intricate than multispectral and RGB sensors because its radiometric and atmospheric calibration workflows are more involuted [40]. Therefore, several steps were required for the hyperspectral imaging processing procedure in order to obtain precise output [33]. The processing of hyperspectral imaging signifies the utilisation of computer algorithms. It includes tasks such as extracting, storing and falsifying information from visible near-infrared (VNIR) or near-infrared (NIR) hyperspectral images. It also provides different information on processing and data mining assignments (for example, analyse, classify, target detection, regression, and pattern identification) [41,42]. Hyperspectral imaging includes extensive data collection stored in pixels while each data particularly correlates to their neighbours [43]. Hyperspectral imaging also comprises the spectral-domain signal as each of the image pixels contains the spectral information; thus, specific tools and approaches have been amplified for processing both spatial and spectral information [42]. This magnitude of data has led to the integration of chemometric and visualisation equipment to competently mine for significant and detailed information [11]. The ordinary hyperspectral image preprocessing procedure is delineated in Figure 2 below [42].

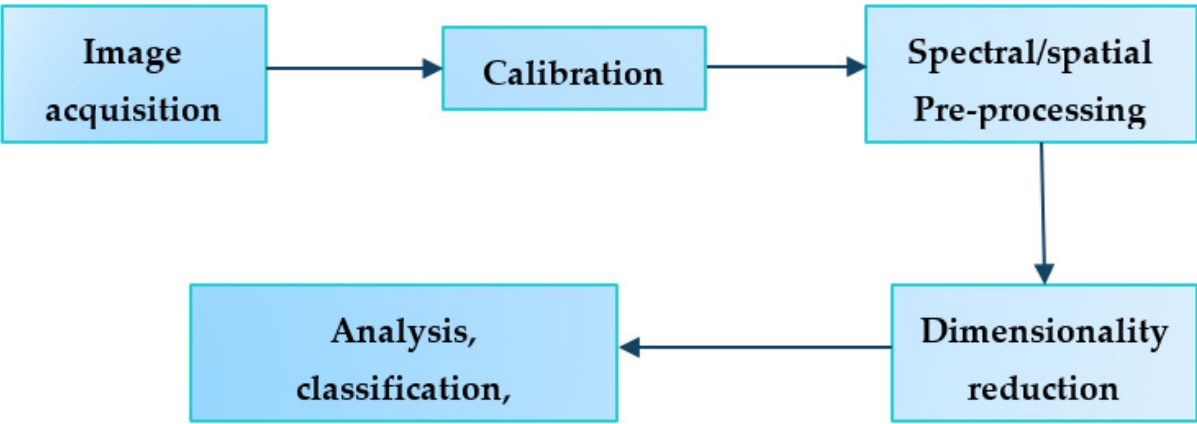

**Figure 2.** Hyperspectral image preprocessing workflow [42].

According to Burger and Geladi [44], numerous amounts of raw data produced from hyperspectral imaging devices contain lots of errors that can be rectified by calibration. Spatial calibration is one of the steps that correlates each image pixel to known units or features, bestowing information about the spatial dimensions and also rectifying the optical aberrations (smile and keystone effects) [42]. However, three conditions could prevail which invalidate calibration models which are: (1) chemical or physical substitution in samples, (2) change of equipment due to inherent uncertainty or ageing parts and, (3) environment/weather condition, for example, temperature or humidity [14]. Lu et al. [36] mentioned that hundreds of bands are common in hyperspectral photographs, and many of them are highly connected. As a result, dimension reduction is an important step to consider while pre-processing hyperspectral images. Dimensionality reduction is a crucial pre-processing step in hyperspectral image classification that reduces HSI's spectral

redundancy, resulting in faster processing and higher classification accuracy. Methods for reducing dimensionality convert high-dimensional data into a low-dimensional space while keeping spectral information [45]. Hence, pre-processing is an important step in increasing the quality of hyperspectral images and preparing them for subsequent analysis.

Basantia et al. [46] stated that hyperspectral imaging generates extensive data collection from a single sample and with thousands of samples that require daily analysis. According to Tamilarasi and Prabu [47], in contrast to other statistical techniques, hyperspectral image analysis uses physical and biological models to absorb light at certain wavelengths. For example, air gases and aerosols could absorb light at specific wavelengths. Dispersion (adding an outside light source to the sensor region of perspective) and absorption are examples of atmospheric diminution (radiance denial). As the outcome, a hyperspectral sensor could not differentiate the radiance recorded with the imaging generated at other times or locations. Hyperspectral image analysis techniques are derived from spectroscopy, which relates to the distinct absorption or patterns of reflection of the context at different wavelengths of a certain material's molecular composition. This image must be subjected to appropriate atmospheric correction techniques in order to compare each pixel's reflection signature to the spectrum of known material; in laboratories and in "library" storage areas, known spectral information of materials include soils, minerals and vegetation types.

*5.2. Hyperspectral Image Classification*

Hyperspectral imaging (HSI) is classified as supervised, unsupervised, and semi-supervised based on the nature of available training samples. The supervised technique uses ground truth information (labelled data) for classification whereas the unsupervised technique does not require any prior information [48]. According to Wenjing and Xiaofei [39], support vector machines, artificial neural networks, decision trees and maximum likelihood classification methods are examples of commonly used supervised classification methods. The basic process is to first determine the discriminant criteria based on the known sample category and prior knowledge and then calculate the discriminant function. Therefore, in supervised classification, Freitas et al. [49] stated that support vector machines can produce results that are similar to neural networks but at a lower computing cost and faster rate, making them ideal for hyperspectral data analysis.

Unsupervised classification refers to categorization based on hyperspectral data spectral similarity, for example, clustering without prior knowledge. As stated by Wenjing and Xiaofei [39], unsupervised classification can only assume beginning parameters, build clusters through pre-classification processing, and then iterate until the relevant parameters reach the permitted range since no prior knowledge is employed. Examples of unsupervised classification are K-means classification and the iterative self-organizing method (ISODATA). Lastly, is the semi-supervised classification which trains the classifier using both labelled and unlabelled data. The semi-supervised learning paradigm has been successfully utilized beyond hyperspectral imaging [50]. It compensates for the lack of both unsupervised and supervised learning opportunities. On the feature space, this classification approach uses the same type of labelled and unlabelled data. Because a large number of unlabelled examples may better explain the overall properties of the data, the classifier trained using these two samples has superior generalisation. Examples of semi-supervised classification are Laplacian support vector machine (LapSVM) and self-training [39].

Therefore, hyperspectral imaging can be one of the potential techniques for automatic discriminations between crops and weeds. These sensing technologies have been utilized in smart agriculture and made substantial progress by generating large amounts of data from the fields. Machine learning modelling integrating features has also accomplished reasonable accuracy in order to identify whether a plant is a weed or a crop. Table 5 shows the application of hyperspectral imaging for the discrimination of crops from weeds by using machine learning.

**Table 5.** Hyperspectral imaging for discrimination of crops from weeds using machine learning (adopted from Su. [51]).

| No. | Crop | Weed | Model | Optimal Accuracy | Reference |
|-----|------|------|-------|------------------|-----------|
| 1. | Rice | Barnyard grass, weedy rice | RF, SVM | 100% | Zhang et al. (2019) |
| 2. | Maize | Caltrop, curly dock, barnyard grass, ipomoea spp., polymeria spp. | SVM, LDA | >98.35% | Wendel et al. (2016) |
| 3. | Soybean, cotton | Ryegrass | LDA | >90% | Huang et al. (2016) |
| 4. | Wheat | Broadleaf weeds, grass weeds | PLSDA | 85% | Hermann et al. (2013) |
| 5. | Broadbean, wheat | Cruciferous weeds | ANN | 100% | De Castro et al. (2012) |
| 6. | Sugar beet | Wild buckwheat, Field Horsetail, Green foxtail, Chickweed | LDA | 97.3% | Okamoto et al. (2007) |
| 7. | Wheat | Musk thistle | SVM | 91% | Mirik et al. (2013) |
| 8. | Maize | *C. arvenis* | RF | >90% | Gao et al. (2018) |

RF—random forest; SVM—support vector machines; LDA—linear discriminant analysis; ANN—artificial neural network; PLSDA—partial least square discriminant analysis.

## 6. HRSI Application in Weed Detection Analysis

### 6.1. Weed Classification Using the Spectral Reflectance

Weed classification is important in precision farming because weeds are pests to crops and compete for space, nutrients, water, and light, and obstruct the growth of crops in the field [52]. Effective weed management is vital in smart agriculture as weeds can trigger major environmental and economic problems in agriculture [53]. According to Su [51], smart agriculture may utilise intelligent technology to precisely monitor weed dispersion in the field and undertake weed control chores at specific locations, which not only improves pesticide effectiveness but also increases the economic benefits of agricultural products. The most significant aspect of an automatic weed removal system within crop rows is the use of dependable sensing technology to accomplish accurate weed and crop discrimination at specified points in the field. Therefore, the application of remote sensing employed in agricultural research was established for the interaction between electromagnetic radiation and plant materials on the Earth's surface [54,55]. Hyperspectral imaging has been suggested as the most suitable instrument for food quality assessment and safety investigation that has been exerted on an array of spectral imaging modalities, for example, NIR, fluorescence, and Raman hyperspectral imaging [46,56]. Hyperspectral images captured by UAV platforms has lately emerged as a significant tool in agricultural remote sensing, with considerable potential for weed detection and species differentiation [57].

To obtain detailed spectrum information, hyperspectral imaging sensors frequently use more and narrower bands. Hyperspectral images have comprehensive spectrum information in each pixel, which has been used for a number of agricultural applications [37]. According to Pott et al. [58], spectral bands can be utilized for differentiating plants from other non-targets. Plant pigments, such as chlorophyll (chlorophyll a and b), carotenes and xanthophylls, are primarily affected by visible light reflection in plant leaves and canopies [35]. The red-edge band reflectance is affected by a mixture of chlorophyll, intense light scattering and internal cellular plant structure. Internal leaf structure and many leaf layers influence the reflectance qualities of the canopy in the near-infrared (NIR) band [37,58].

Paap [19] stated that plants' spectral reflectance is identified based on the cellular and biochemical leaf structure and leaf canopy. Figure 3 represents a typical spectrum reflectance and transmitted wavelength of green leaf. The contrast of the reflectance and transmittance spectra depend upon absorption which is in the visible spectrum range 400–700 nm, the spectra are controlled by absorption of various pigments and primarily chlorophylls. In near infra-red (NIR), the reflectance spectra are high which is close to 50% and flat while above 1300 nm, the reflectance declines because of the water absorption present in the leaf.

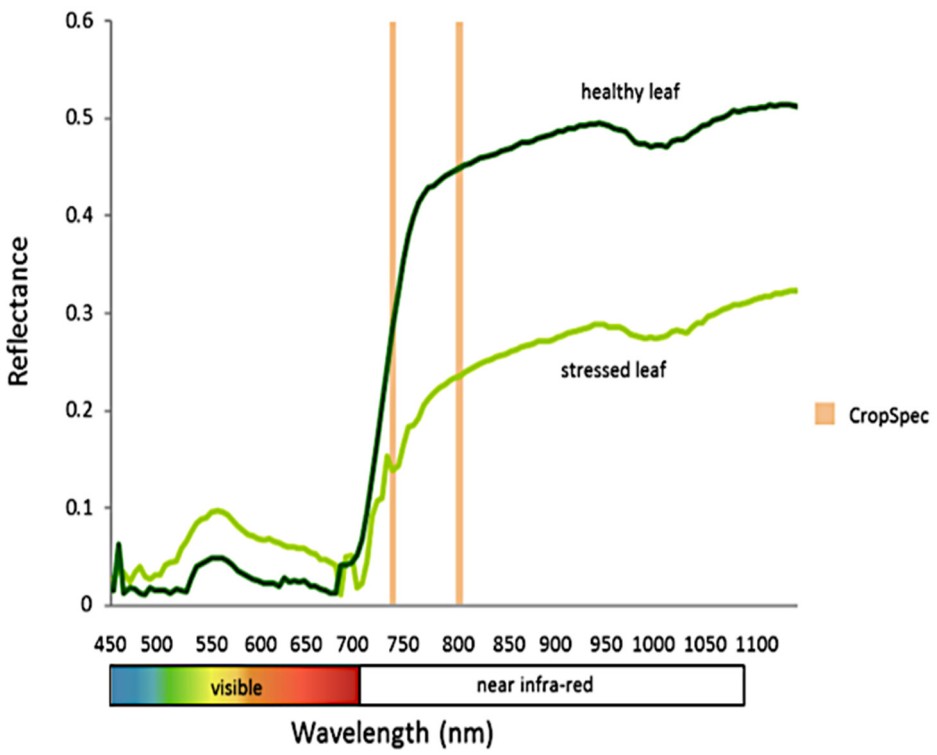

**Figure 3.** Spectral reflectance for healthy and stressed leaf in visible and NIR wavelength [59].

Further contemplation in the vegetation mapping procedure is the size of the objects to be mapped. Higher spatial resolution imagery is frequently used for mapping narrow vegetation objects which are acquired from airborne sensors [60]. Weeds compete with crops and are difficult to distinguish because of their similar colour, shape, and size [61]. However, a previous study on dispersing reflectance spectra of crop and weed leaves found the potential of weed detection with reflectance measurements. Zhang et al. [62] mentioned that due to the considerable absorption by chlorophylls, a plant leaf typically has a low reflectance in the visible spectral range and a comparatively high reflectance in the near-infrared spectral area due to internal leaf scattering and no absorption. Therefore, according to Thenkabail et al. [63], plant leaf area index and biomass are more sensitive to the red band at roughly 680 nm, while plant moisture status is more sensitive to the NIR near 950 nm. The correlation among the plant pathology has been employed by the remote sensing technique in contemplation to discover the discrete plant characteristics from their spectra reflectance.

The application of hyperspectral remote sensing is comprehensively used for different weed detection analysis studies, for example, weed discrimination in maize [64], discrimination of grassweeds in winter cereal crops [65], in early detection of spotted knapweed (*Centraurea maculosa*) and babysbreath (*Gypsophila paniculate*) with hyperspectral sensor [66], herbicide-resistant weeds classification [67], identification of (*Ranunculus acris* [giant buttercup] and *Cirsium arvense* [Californian thistle]) by Li et al. [53] and spectral features extraction from hyperspectral images to differentiate weedy rice and barnyard

grass [68]. Singh et al. [57] stated that hyperspectral imaging application has also been used to identify between crop types, for instance, the utilization of satellite-based hyperspectral sensors to distinguish mustard, potato, sugarcane, sorghum, and wheat in the range of 700–750 nm. The efficacy of hyperspectral sensors for plant species characterization has been documented in a number of different studies, which is included in Table 6 below.

**Table 6.** Published reports/study on crop and weed species classification using hyperspectral imagery [57].

| Crop/Weed Mixture | Reflectance | Wavelength (nm) |
|---|---|---|
| **Wheat system** | | |
| Wheat (*Triticum aestivum*) | 0.42–0.45 | 720–850 |
| Broadleaved weeds | 0.6–0.8 | 720–850 |
| Grass weeds | 0.55–0.6 | 720–850 |
| Wheat stubbles | 0.2–0.3 | 1425–2250 |
| Wheat stubbles heavily grazed | 0.3–0.4 | 1425–2250 |
| Cruciferous weeds | 0.65–0.7 | 750–900 |
| Wheat | 0.4–0.5 | 720–900 |
| **Soybean system** | | |
| Soybean (*Glycine max*) | 0.85–0.90 | 750–900 |
| Prickly sida (*Sida spinosa*) | 0.78–0.82 | 750–900 |
| Pitted morning glory (*Ipomoea lacunosa*) | 0.6–0.65 | 750–900 |
| Sicklepod (*Cassia obtusifolia*) | 0.52–0.55 | 750–900 |
| **Cotton system** | | |
| Cotton (*Gossypium hirsutum*) | 0.48–0.52 | 400–600 |
| Cogongrass (*Imperata cylindrica*) | 0.5–0.6 | 400–750 |
| Johnsongrass (*Sorghum halepense*) | 0.45–0.52 | 400–750 |
| Sicklepod (*C. obtusifolia*) | 0.21–0.3 | 400–800 |
| **Sorghum system** | | |
| Sorghum (*Sorghum bicolor*) | 0.55–0.6 | 720–1000 |
| Common lambsquarters (*Chenopodium album*) | 0.39–0.41 | 720–1000 |
| Pigweed (*Amaranthus* spp.) | 0.48–0.52 | 720–1000 |
| Barnyardgrass (*Echinochloa* spp.) | 0.35 | 72–1000 |
| Mallow (*Malva* spp.) | 0.42–0.44 | 720–1000 |
| Purple nutsedge (*Cyperus rotundus*) | 0.28–0.30 | 720–1000 |

*6.2. Algorithms and Modelling for Weed Detection Analysis*

For various agricultural applications, several remote sensing approaches, such as hyperspectral data from airborne, satellite platforms using multispectral and optical imagery have been proposed [69,70]. A Study conducted by Felegari et al. [71] looked into the drawbacks and benefits of using a combination of radar data and optical images to determine the types of crops in the Tarom region (Iran) in which the Sentinel 1 and Sentinel 2 images were utilised to generate a map for the selected research area. Hyperspectral sensing, which measures reflectance from visible to shortwave infrared wavelengths, has allowed vegetation to be classified and mapped at a variety of taxonomic scales, often down to the species level. To reduce the dimensionality of the data to a level suitable for the creation of a classification model, hyperspectral measurements recorded by narrowband spectroradiometers or imaging sensors have typically required some type of spectral feature selection [72]. Therefore, the remote sensing method can detect the existence of non-crop plants between rows, such as the recognition of weeds within rows, whereas segregating weeds from crops and identifying weed species emerging from proximal sensing research has utilized both spectral reflectance and leaf shape analysis for identification [73]. According to Lan et al. [74], to examine the datasets generated by these methodologies, proper and effective sophisticated algorithms as well as high-power computation are required. Genetic programming was utilized by Nguyen et al. [75] to distinguish between rice and other leaf groups. They also employed a scanning window of 20 × 20 pixels on a test image to evaluate the classifier,

attaining a 90% accuracy by applying the classifier to each pixel of the window based on a colour threshold.

Therefore, several automatic classification techniques have been employed to classify remote sensing data and plant monitoring procedures, for example, the machine learning method [74]. According to Dadashzadeh et al. [18], machine vision based on image processing has been used to collect data in two different ways: Two-dimensional (2D) vision and three-dimensional (3D) vision. When using 2D cameras, machine vision systems based on two-dimensional (2D) image processing have some drawbacks. First, differences in external illumination have an impact on the quality of images captured by 2D cameras; thus, the camera's field of view must be covered. Second, the overlap of different plant components can make distinguishing weeds from crops difficult.

According to Perez-Ortiz et al. [76], most standard classifiers in machine learning are based on learning a discriminant function from labelled data (i.e., supervised learning). However, obtaining tagged data, as opposed to unlabelled data, can be time consuming and costly. Liakos et al. [77] stated that machine learning (ML) has risen to prominence with big data technology and high-performance computers to open up new avenues for unravelling, quantifying, and understanding data-intensive processes in agricultural operations. ML is characterised as a scientific subject that allows machines to learn without being strictly programmed, among other things. Examples of ML modelling include artificial neural networks (ANNs), Bayesian models (BM), deep learning (DL), dimensionality reduction (DR), decision trees (DT), ensemble learning (EL), instance-based models (IMB) and support vector machines (SVMs).

Dadashzadeh et al. [18] investigated site-specific weed management in the rice field using two metaheuristic algorithms: The bee algorithm (BA) and particle swarm optimisation (PSO), in order to improve the neural network's ability to identify the most effective characteristics and classify different types of weeds. Because of their abundance in the chosen region, this study focused on a rice cultivar (*Tarom Mahali*) and two common types of weeds (narrow-leaf weeds (*Echinochloa crus-Galli*, *Paspalum distichum*, and *Cyperus difformis*) and wide-leaf weeds (*Alisma plantago-aquatica* and *Eclipta prostrata*) while a stereo camera was used to collect the necessary data in the form of stereo videos, with different channels of each frame extracted. The proposed stereo vision technique, which averaged the related points on various channels and the proposed hybrid ANN-BA classifier for better classification accuracy, proved to have promising capabilities. Zheng et al. [78] created and evaluated a new classification algorithm based on colour indices and support vector data description (SVDD). In the first, second and third years of a three-year case study, overall accuracies of 90.19%, 92.36%, and 93.8%, respectively, were achieved. Kamath et al. [79] looked at how to categorize paddy crops and weeds from digital images utilizing several classifier systems developed with support vector machines (SVM) and random forest classifiers (RFs) in which the dataset included paddy plants and weeds from the seedling stage (1-leaf seedling) to the flowering stage. The results with an accuracy of 91.36% showed that multiple classifier systems were shown to outperform single classifier systems and the extracted features are good for paddy crops and weeds classification.

Li et al. [53] studied weed identification by using hyperspectral data images trained on three classification models, namely partial least squares discriminant analysis, support vector machine and multilayer perceptron (MLP) with an overall accuracy range of about 70–100%. The analysis was run by using the whole plant averaged (Av) spectra and superpixels (Sp) averaged spectra from four different weed samples which comprised two types of grass (*Setaria pumila* [yellow bristle grass] and *Stipa arundinacea* [wind grass]) and two broadleaf weed species (*Ranunculus acris* [giant buttercup] and *Cirsium arvense* [Californian thistle]). Results showed that using both Av and Sp spectra were able to identify the four weeds' species. To solve the challenge of forecasting the pre-planting risk of Stagonospora nodorum blotch (SNB) in winter wheat, Mehra et al. [80] used machine learning approaches such as artificial neural networks (ANNs), category and regression

trees, and random forests (RFs). They created risk assessment models that could help with disease control decisions before planting the wheat crop.

Research conducted by Chen et al. [81] combined the application of multi-feature fusion and support vector machine (SVM) to detect corn seedlings and weeds for limiting crop damage with an average recognition accuracy of about 97.50%. The dataset included a small database of corn seedlings and weed and actual field images. The results of the experiments revealed that the fusion feature of rotation invariant local binary pattern (LBP) feature and grey level-gradient co-occurrence matrix based on an SVM classifier accurately detected all types of weeds and corn seedlings. This provides information about weed and crop positions to the spraying herbicide, allowing for exact spraying and fertilising. Chou et al. [82] used a wavelet packet transform paired with a weighted Bayesian distance based on crop texture and leaf data to identify the crop. The dataset needed for this study included field crop images captured with a digital camera with a resolution of $640 \times 480$ pixels. To discriminate plants, they estimated energy coefficients in multiple frequency bands produced after the change. The crop identification achieved an accuracy of 94.63% by using the decision distance in different climates over three consecutive days of photography.

Bakhshipour and Zareiforoush [83] used integrate decision tree (DT) and fuzzy logic techniques to establish a fuzzy model for differentiating the peanut plant from broadleaf weeds with the overall accuracies on training and testing datasets being, respectively, 92% to 96%. On the input dataset, two feature selection approaches were utilised: Principal component analysis (PCA) and correlation-based feature selection (CFS), and three decision trees (DTs) were used to distinguish between distinct plants: J48, random tree (RT), and reduced error pruning (REP). Another study by Bakhshipour et al. [84] is on texture features recovered from wavelet sub-images to detect and describe four species of weeds in a sugar beet field, while neural networks (NN) were run as a classifier. Images were taken from sugar beet fields with a resolution of $96 \times 1280$ pixels when the plants had six to eight leaves with significant occlusion and a height of about 80 mm to 160 mm. The research found that even at a stage of beet growth greater than six leaves, the application of wavelets proved to be effective for weed detection. Two-dimensional Gabor filters were employed to extract the features in a study conducted by Tang et al. [85], and an artificial neural network (ANN) was utilized to categorize broadleaf and grass weeds. The seeds of the selected broadleaf weed species were planted and the image was captured four weeks after seeding. The Gabor wavelet/ANN system was created to use texture features to classify weed images into broadleaf and grass categories. Their findings revealed that joint space–frequency texture properties might be used to classify weeds.

Furthermore, in agricultural research, deep learning combined with advancements in computer technology, particularly graphical processing units (GPU) embedded processors, has produced remarkable results for image classification and objection detection [86,87]. According to Alom et al. [88], deep learning (DL) algorithms have many advantages over traditional machine learning approaches for image classification, object detection and localization. To build a feature extractor from raw data, traditional machine learning techniques necessitate extensive domain knowledge [89,90]. The DL approach, on the other hand, employs a representation-learning method in which a machine can automatically discover discriminative features from raw data for classification or object detection problems. DL methods can effectively extract discriminative features of crops and weeds due to their strong feature learning capabilities. Furthermore, as data sets have grown larger, the performance of traditional machine learning approaches has become saturated. When large datasets are used, DL techniques outperform traditional machine learning techniques [88].

Hosseini et al. [91] stated that convolutional neural networks (CNNs) and recurrent neural networks (RNNs) are two commonly used architectures in DL. Although CNNs are used for other types of data, the most common application of CNNs is to analyse and classify images. The term convolution refers to the filtering process. CNN is based on a stack of convolutional layers. Each layer receives input data, transforms or convolves

it, and outputs it to the next layer. This convolutional operation eventually simplifies the data so that it can be processed and understood more easily [89]. Mentioned by Bah et al. [92], convolutional neural networks (CNNs) have advanced primarily as a result of their successful use as a method in the ImageNet Large-Scale Vision Recognition Challenge 2012 (ILSCVR12) and the creation of the AlexNet network in 2012, which demonstrated that a large, deep convolutional neural network can achieve record-breaking results on a highly challenging dataset using purely supervised training. Therefore, a deep convolutional neural network (DCNN) system for plant recognition based on plant leaf features and patterns was also documented by Lee et al. [93] based on the leaf's shape, texture, and venation while they presented new hybrid models taking advantage of the correspondence of different contextual information of leaf features.

## 7. Direction for Future Work and Conclusions

In most situations, removing weeds in agricultural areas requires the use of large amounts of chemical pesticides, which are damaging to the environment regardless of how effective they are at enhancing crop output. Precision spraying might be explored to optimise herbicide application in crop fields, thanks to recent advances in image sensors. In this paper, we have reviewed the situation of weeds in rice crops, the background of hyperspectral imaging and techniques for processing hyperspectral data. This study is interdisciplinary and experts from various disciplines, such as agronomy (weed science), remote sensing, computing, and engineering are collaborating. Hyperspectral remote sensing technology is an important component in precision farming and is being used by a growing number of scientists and agricultural researchers. The capacity to properly and reliably distinguish weeds from crops is a vital step in controlling or eradicating weed infestations in agricultural crops. Due to the abundance of spectral information sensitive to distinct plant biophysical and biochemical properties, hyperspectral imaging offers a lot of potential for applications in agriculture, especially precision agriculture. Hyperspectral remote sensing technology uses the difference in spectral reflectance qualities between weeds and crops to identify weeds in crop stands and aids in the compilation of weed maps in the field, allowing for the application of site-specific and need-based herbicides for weed management.

Hyperspectral imaging data with high spatial resolution along with machine learning algorithms in remote sensing showed good potential in agricultural studies. In the recent decade, sensing technologies and machine learning approaches have grown at a breakneck pace. These advancements are expected to continue to provide more cost-effective and comprehensive datasets, as well as more advanced algorithmic solutions, allowing for better crop and environment status estimates and decision making. For more intricate hyperspectral picture classification, existing theories and algorithms still have some limitations. As a result, future research efforts will focus on developing more tailored hyperspectral image classification systems. Therefore, in order to successfully use the information on weeds and crop monitoring for economic benefit, a state or district level information system based on existing information on diverse crops produced from this hyperspectral remote sensing approach is required. Governments can use hyperspectral remote sensing data to make critical decisions about which policies to pursue and how to address agricultural concerns.

**Author Contributions:** Conceptualization, N.S. and N.N.C.; methodology, N.S. and M.H.M.R.; resources, N.S.; writing—original draft preparation, N.S.; writing—review and editing, N.S., N.N.C., A.S.J., N.M.N., W.F.F.I. and M.H.M.R.; visualization, N.S. and N.N.C.; supervision, N.N.C., N.M.N., M.H.M.R. and A.S.J.; project administration, A.S.J.; funding acquisition, A.S.J. All authors have read and agreed to the published version of the manuscript.

**Funding:** This research was funded by Research Project entitled "Pest and Disease Monitoring Using Artificial Intelligent for Risk Management of Rice Under Climate Change" under the Long-term Research Grant Scheme (LRGS), Ministry of Higher Education, Malaysia, LRGS/1/2019/UPM/01/2/5 (vote number: 5545002).

**Institutional Review Board Statement:** Not applicable.

**Informed Consent Statement:** Not applicable.

**Data Availability Statement:** Not applicable.

**Acknowledgments:** The authors would like to thankfully acknowledge the research project "Pest and Disease Monitoring Using Artificial Intelligent for Risk Management of Rice Under Climate Change" under Long-term Research Grant Scheme (LRGS), Ministry of Higher Education Malaysia for providing financial support, LRGS/1/2019/UPM/01/2/5 (Vote: 5545002).

**Conflicts of Interest:** The authors declare no conflict of interest.

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
