# Peer review of "The Application of Hyperspectral Remote Sensing Imagery (HRSI) for Weed Detection Analysis in Rice Fields: A Review"

_applsci, doi:10.3390/app12052570_

Round 1

Reviewer 1 Report

The paper presents an overview on the Application of Hyperspectral Remote Sensing Imagery (HRSI) for weed detection analysis in rice fields. The review is well conceived and organized. However, the final part of the paper should (Section 4.2 Algorithms and Modelling for Weed Detection Analysis) probably be more comprehensive and informative on achieved results (as it present the so far achieved results in observed area), and if (and when) possible some comparison of proposed methods should be given in the context of detection success, complexity, data needed , ...

Also, the English language should be improved.

Reviewer 2 Report

Weeds are quite popular to be founded in the field and they are competed for light, water, and etc. with the crops. For the modern agriculture, Hyperspectral Remote Sensing Imagery (HRSI) is good for weed detection. On other hand deep learning is also good for modern agriculture. For the review paper, it seems better to include some deep learning neural networks papers for weed detection.

Round 2

Reviewer 1 Report

The paper presents an overview on the Application of Hyperspectral Remote Sensing Imagery (HRSI) for weed detection analysis in rice fields. The review is well conceived and organized.

In the revised version, authors included changes which are in line with the issues reported for the previous version of the manuscript.
